# HnRNP Pathologies in Frontotemporal Lobar Degeneration

**DOI:** 10.3390/cells12121633

**Published:** 2023-06-15

**Authors:** Xinwa Jiang, Ariana Gatt, Tammaryn Lashley

**Affiliations:** 1Queen Square Brain Bank for Neurological Disorders, UCL Queen Square Institute of Neurology, London WC1N 1PJ, UK; xinwa.jiang.21@ucl.ac.uk; 2Department of Neurodegenerative Disease, UCL Queen Square Institute of Neurology, London WC1N 3BG, UK

**Keywords:** frontotemporal lobar degeneration, FTLD, heterogeneous ribonucleic acid protein, hnRNP, frontotemporal dementia, FTD

## Abstract

Frontotemporal dementia (FTD) is the second most common form of young-onset (<65 years) dementia. Clinically, it primarily manifests as a disorder of behavioural, executive, and/or language functions. Pathologically, frontotemporal lobar degeneration (FTLD) is the predominant cause of FTD. FTLD is a proteinopathy, and the main pathological proteins identified so far are tau, TAR DNA-binding protein 43 (TDP-43), and fused in sarcoma (FUS). As TDP-43 and FUS are members of the heterogeneous ribonucleic acid protein (hnRNP) family, many studies in recent years have expanded the research on the relationship between other hnRNPs and FTLD pathology. Indeed, these studies provide evidence for an association between hnRNP abnormalities and FTLD. In particular, several studies have shown that multiple hnRNPs may exhibit nuclear depletion and cytoplasmic mislocalisation within neurons in FTLD cases. However, due to the diversity and complex association of hnRNPs, most studies are still at the stage of histological discovery of different hnRNP abnormalities in FTLD. We herein review the latest studies relating hnRNPs to FTLD. Together, these studies outline an important role of multiple hnRNPs in the pathogenesis of FTLD and suggest that future research into FTLD should include the whole spectrum of this protein family.

## 1. Frontotemporal Dementia and Frontotemporal Lobar Degeneration

Frontotemporal dementia (FTD) is an umbrella term for several clinical symptoms that are characterised by cognitive impairments related to the frontal and temporal lobes, such as behavioural and personality changes and language impairments [1]. FTD is the second most common young-onset dementia (<65 years), following familial Alzheimer’s disease [2,3]. FTD can be divided into three main subtypes based on clinical presentation, namely behavioural variant FTD (bvFTD), progressive non-fluent aphasia (PNFA) (or non-fluent variant primary progressive aphasia, nfvPPA), and semantic dementia (SD) (or semantic variant of primary progressive aphasia, svPPA) [4]. Among them, bvFTD is the subtype with the highest prevalence, accounting for approximately 75% of FTD cases [5]. The symptoms of bvFTD are characterised mainly by changes in behaviour and personality, as well as impairments of executive functions. Although some cases may also exhibit some language impairments, particularly in the late stages of the disease, these are generally more subtle [6]. The other two subtypes of FTD both present primarily with symptoms of language impairments, but they differ in their clinical presentation. PNFA patients are mainly characterised by the effortful production of speech, as well as disfluent expressions and incorrect grammatical usage, but the patient usually retains a normal comprehension of word meanings [7]. On the other hand, SD is mainly manifested by the patient’s loss of the concept of words. Although patients can usually produce language fluently, their speech often appears empty of meaning [8]. In addition, around 10–15% of FTD cases are accompanied by motor neuron disease (MND) in the form of amyotrophic lateral sclerosis (ALS) [6,9]. This FTD-MND type is usually present in the bvFTD subtype, which exhibits a more rapid disease progression and an increase in the severity of symptoms [10]. As the onset of FTD is usually early, sometimes even before the retirement age, it often has a huge impact on the patient’s life, work, and social activities. This also results in FTD imposing a huge financial and emotional burden on the patient’s family, as well as society. Therefore, the study of FTD pathogenesis and the search for potential prevention or treatment options are of important medical and social value.

Pathologically, the predominant cause of FTD is frontotemporal lobar degeneration (FTLD). FTLD is not a clinical condition, but a pathological term for this neurodegenerative disease that causes progressive neuronal death in the frontal and temporal lobes [11]. A high proportion of FTLD is familial, caused by mutations in the following genes: *C9orf72*, *MAPT*, *GRN*, *FUS*, *VCP*, *Ataxin 2*, *Chmp2B*, *TMEM106B*, *OPTN*, and *PFN* [12]. Among them, an excessive extension of the *C9orf72* GGGGCC repeats (also known as hexanucleotide repeat expansions, HREs) is the most common genetic factor causing FTLD, which accounts for around 25% of familial FTLD cases and even some sporadic cases [13]. GGGGCC repeats in healthy populations generally do not exceed 26, but in *C9orf72*-associated FTLD cases, the number of repeats can be as high as 60–1600 [13,14,15,16]. The HREs impair the normal expression of the *C9orf72* gene and lead to the loss of C9orf72 function [17]. Moreover, these HREs can also be transcribed into both sense and anti-sense RNA sequences, which in turn can accumulate to form RNA foci. These RNA foci segregate a range of RNA-binding proteins, leading to disruptions in the transcriptional regulation and splicing of other RNAs, which in turn leads to an aberrant expression of many proteins. This process is also known as RNA toxicity [18]. In addition, these sense and anti-sense RNAs may be translated via the repeat-associated non-ATG (RAN) translation pathway to form a number of toxic dipeptide repeats (DPRs), such as polyGA, polyGP, polyGR, polyPR, and polyPA [18]. Together, these mechanisms lead to neurotoxicity and result in neurodegeneration.

As with most neurodegenerative diseases, FTLD is a proteinopathy in which neuronal death is triggered by the aggregation of misfolded proteins. The three proteins that have been extensively studied in relation to FTLD are tau, TAR DNA-binding protein 43 (TDP-43), and fused in sarcoma (FUS) [19]. They have been found to aggregate and form inclusions in different cases of FTLD. Therefore, FTLD can also be classified into the sub-types FTLD-tau, FTLD-TDP, and FTLD-FUS. In addition, FTLD-TDP cases have been further classified into subtypes FTLD-TDP A–E, as they exhibit distinct lesion types and the distribution of pathological inclusions [20]. Among these proteins, TDP-43 and FUS both belong to the heterogeneous nuclear ribonucleoproteins (hnRNPs) family [21]. In particular, the aggregation of TDP-43 and the formation of inclusions are the main pathological features in approximately 50% of FTLD and 97% of ALS cases [22,23]. In the formation of neuronal cytoplasmic inclusions, TDP-43 is lost from the nucleus and has been found to lead to a number of RNA splicing abnormalities, affecting an important neuronal growth-related factor gene, *STMN2* [24,25], as well as a gene critical for the synaptic function of *UNC13a* [26,27]. Nuclear depletion of TDP-43 leads to aberrant splicing in both *STMN2* and *UNC13a* mRNAs, incorporating a cryptic exon leading to reduced protein levels, which negatively affect neuronal function [24,25,26,27]. In addition to TDP-43, other hnRNPs have been found to have similar cryptic exon suppression functions, such as hnRNP C, I, L, M, and K [28,29,30,31,32], although not all of them have been shown to be directly related to FTLD as of yet. In addition, the autoregulation of many hnRNPs through negative feedback has also been suggested to possibly lead to a reduction in the expression of other functional proteins [21]. Therefore, from both genetic and pathological perspectives, it is implied that there may be other hnRNPs also involved in the pathogenesis of FTLD. Indeed, in recent years, there has been a growing number of pathological and mechanistic studies on hnRNPs in FTLD. The aim of this review is to provide an overview and summary of the findings on hnRNP pathologies in FTLD spanning recent years. As TDP-43 and FUS have been extensively studied and reviewed in many publications elsewhere, this review aims to focus mainly on hnRNPs other than TDP-43 and FUS.

## 2. Heterogeneous Nuclear Ribonucleoproteins

hnRNPs are a class of RNA-binding proteins (RBPs) which share many structural and functional similarities [33]. More than 20 hnRNPs with different structures have been identified and named alphabetically (A to U) [34]. Some hnRNPs were identified and named before the definition of hnRNP was established, such as FUS (hnRNP P), so their original nomenclature by convention usually remains. More details of the nomenclature have been discussed in a previous review [21] and are therefore not repeated here.

Structurally, hnRNPs typically contain several core functional domains responsible for RNA recognition and binding, nuclear localisation, and nucleocytoplasmic shuttling, respectively. Among these, RNA recognition and binding domains typically include RNA-recognition motifs (RRMs), glycine–arginine-rich (GAR) domains (known as the Arg-Gly-Gly repeat domain, RGG boxes), and K-homology (KH) domains, but their presence and distribution vary among different hnRNPs [21]. For example, the KH domains are present only in hnRNP E1, E2, and K. Some hnRNPs may also contain DNA-binding domains, such as the zinc finger domain in hnRNP P (FUS) and SAP in hnRNP U [21]. In addition, many hnRNP structures contain low complexity domains (LCDs) or intrinsically disordered regions (IDRs). These regions have a high degree of amino acid repeats and low complexity, and may therefore form higher-order ribonucleoprotein complexes and interact with each other, leading to LCD-driven liquid–liquid phase separation (LLPS) [35,36]. LLPS then has the potential to prompt other hnRNPs to undergo separation, which exhibits a prion-like seeding effect [37].

On the other hand, the nuclear localisation/transport domains in hnRNPs include the nuclear localisation signal (NLS) or M9, which has a bi-directional function of nuclear import and export [38]. It is because of these domains that hnRNPs generally exhibit predominantly nuclear localisation and may be accompanied by a lesser degree of cytoplasmic localisation. However, not all hnRNPs contain nucleus localisation/transport-associated domains. Among them, hnRNP A1 and A2B1 are relatively typical examples, as one of their features is their inclusion of the M9 domains. In addition, hnRNP A2B1, C, G, Q, and R also contain typical NLS domains [21].

Functionally, hnRNPs are widely expressed in many tissues and can bind to RNAs and other RBPs to form dynamic complexes which participate in all processes of mRNA maturation and metabolism, including splicing, capping, translocation, polyadenylation, and stabilisation [34]. These functions need to be carried out mainly in the nucleus, and therefore hnRNPs are mainly located in the nucleus. However, through the nuclear export signalling domains, some hnRNPs can also shuttle from the nucleus to the cytoplasm and participate in cytoplasmic functions such as mRNA translocation from the nucleus to the cytoplasm and translation [39]. Consequently, hnRNPs have important regulatory roles in normal gene expression, and their dysfunction, nuclear depletion, and cytoplasmic inclusion formation can lead to abnormal transcription and translation of a range of downstream target RNAs. One important pathway of mRNA regulation by hnRNPs is nonsense-mediated mRNA decay (NMD). When hnRNPs bind to introns of target transcripts, they may cause an aberrant exon inclusion (e.g., short exon 6A of hnRNP L [40]) or exon skipping (e.g., exon 11 of hnRNP I [41]). This inclusion and/or skipping may result in a frameshift of mRNA sequences, which may lead to the introduction of premature termination codons (PTCs) [21,42]. Most PTC-containing transcripts can activate the NMD pathway, leading to the degradation of these transcripts and subsequent reduction in the levels of their encoded proteins. The negative feedback autoregulation of many hnRNPs is also dependent on the NMD pathway [21]. Therefore, abnormal mRNA regulation can lead to altered levels of functional proteins or loss of function, for example, nuclear depletion of TDP-43 leads to a decrease in downstream STMN2 expression and consequently neurodegeneration [25]. However, the role of hnRNPs in the pathology of neurodegenerative diseases, particularly FTLD, still lacks comprehensive investigation, leaving many mechanisms still underexplored.

As with many RNA-binding proteins (RBPs), hnRNPs can form granules through LLPS [43]. The maturation of these granules can lead to the formation of amyloid-like fibrous structures, resulting in enriched amyloid aggregates and promoting the development of neurodegenerative diseases such as FTLD [43]. Meanwhile, many hnRNPs may be recruited into stress granules under stress conditions, sequester non-translating mRNAs [22,44,45], and facilitate the formation of RNA foci [45], thereby leading to RNA toxicity in affected neurons. Due to the complexity and diversity of hnRNPs in FTLD pathology, the following section will focus primarily on listing hnRNPs that have been directly associated with FTLD pathology (Table 1). Moreover, as many classical associations of hnRNPs with FTLD have been previously reviewed [21], the following section will place particular emphasis on reviewing and discussing the most recent findings, with the hope to provide some insights into future research avenues.

### 2.1. HnRNP A1 and A2B1

HnRNPA1 and hnRNPA2B1 are highly expressed hnRNPs which can interact directly with TDP-43 to collaboratively regulate RNA metabolism [44]. HnRNP A2B1 can be expressed in two isoforms, A2 and B1, of which A2 is 12 amino acids shorter than B1, but is the predominant isoform, accounting for approximately 90% in tissues [37].

In 2013, pathogenic mutations in hnRNP A1 (c.785/941A.T, p.D262V/D314V) and hnRNP A2B1 (c.869/905A.T, p.D290V/D302V) located at their prion-like domains (PrLDs) (within the LCD sequences) were identified in several multisystemic proteinopathy (MSP) cases [37]. MSP is a rare neurodegenerative disease that affects multiple systems, including muscle, brain, and bone. Patients can present with symptoms of FTD, ALS, and Paget’s disease of the bone (PDB) [64]. These mutations within PrLD can enhance the potency of the steric zipper motifs of these hnRNPs, thus promoting the fibrillization of hnRNP A1 and A2B1, and even other hnRNPs. In addition, these mutations also result in a more rapid recruitment of hnRNP A1 and A2B1 to RNA granules under stress. The formation of RNA granules is a reversible process through LLPS, of which the normal function is to sequester mRNA and other translational machinery under cellular stress to regulate protein synthesis towards survival [65]. However, as the dynamic balance between stress granule formation and breakdown shifts towards formation, it gradually matures and develops amyloid-like fibrillization, which eventually forms insoluble aggregates [66]. In some MSP cases, mutant hnRNP A1 and A2B1 were found to exhibit nuclear depletion and the formation of cytoplasmic inclusions in a proportion (around 10%) of myofibers. These mutated hnRNPs can also cause nuclear depletion and cytoplasmic inclusion formation in Drosophila and mouse muscle [37]. This study was among the first to investigate the role of hnRNP A1, A2B1 and other hnRNPs (other than TDP-43 and FUS) in neurodegenerative diseases. This study, however, did not reveal a direct association between the loss of nuclear hnRNPA1/A2 B1 and FTLD, and the functional abnormalities in neurons were not investigated.

A later study provided more direct evidence for the association between hnRNP A1 and FTLD when hnRNP A1 inclusions were observed in the frontal cortex and entorhinal cortex of FTLD-FUS cases [46]. Indeed, hnRNP A1 and FUS were co-localised in the cytoplasmic inclusions and neuropil threads of these cases. Although hnRNP A1 expression did not show a significant increase (interestingly, hnRNP A2/B1 expression was increased), it exhibited a shift in distribution from the nucleus to the cytoplasm [46]. In addition, a similar cytoplasmic mislocalisation of hnRNP A1 was also found in the motor neurons of amyotrophic lateral sclerosis cases [47] and in the cortical neurons (not specified if the regions were frontal or temporal) of multiple sclerosis cases [67]. These findings suggest that the nuclear depletion and cytoplasmic aggregation of hnRNP A1 may have a pathological role in different neurodegenerative diseases.

Mechanistic studies suggest that TDP-43 appears to have a regulatory role in the pathological effects of hnRNP A1. In cellular models, TDP-43 was found to bind to hnRNP A1 transcripts, and nuclear depletion/cytoplasmic accumulation of TDP-43 was found to regulate alternative splicing of hnRNP A1, leading to an increased level of hnRNP A1B, a more aggregation-prone isoform of hnRNP A1 [47]. At the same time, nuclear depletion and cytoplasmic accumulation of TDP-43 were found to increase cytoplasmic hnRNP A1 and A1B levels without affecting nuclear hnRNP A1 levels [47].

Moreover, another study suggests that the nuclear-to-cytoplasmic translocation of hnRNP A1 under stress may be associated with the regulation of poly(ADP-ribosyl)ation (PARylation) [48]. PARylation is an important protein post-translational modification that is involved in mediating ribonucleoprotein granules and is associated with DNA repair and cell death. HnRNP A1 has a PARylation site at its K298 locus and a PAR-binding motif between its two closely related RRMs near the N-terminus. According to this study, under stress, PARylation of hnRNP A1 at the K298A site is necessary for its translocation from the nucleus to the cytoplasm, while the binding of PAR or PARylated proteins at the PAR-binding site is important for its recruitment of stress granules [48]. In vitro, PAR promotes the co-liquid–liquid phase separation (co-LLPS) of hnRNP A1 and TDP-43, whereas the PAR hydrolysis enzyme, PARG, reduces this co-LLPS, suggesting that the level of PARylation has a modulatory effect on the interaction of hnRNP A1 and TDP-43. More importantly, in a murine motor-neuron-like cell, NSC-34 model, reduced PARylation levels resulted in the reduced cytotoxicity of hnRNP A1 and TDP-43 [48].

These findings have provided some insights into the regulatory mechanisms by which hnRNP A1 translocates to the cytoplasm, is recruited to stress granules, and leads to cytotoxicity. Similarly, the low-complexity domain (LCD) of hnRNP A2 was found to undergo LLPS on its own and induce co-LLPS of TDP-43 and hnRNPA2 through a direct interaction with the TDP-43 helix [68]. Notably, the LCDs are present in the structures of many hnRNPs, and in hnRNP A1 and A2, the LCDs also correspond identically to the prion-like domains [37,69]. Thus, this pathological mechanism of self-LLPS and induction of co-LLPS with other hnRNPs found in hnRNP A1 and A2 LC domains may be widespread across many hnRNPs. However, all the above mechanisms need to be further validated in FTLD patients. Additionally, whether the pathology of other hnRNPs in FTLD is associated with similar mechanisms is worth exploring in future studies.

### 2.2. HnRNP A3

Unlike hnRNP A1 and A2, hnRNP A3 pathology has been thus far restricted to cases with *C9orf72* mutations. This may be because hnRNP A3 specifically binds to the GGGGCC repeat RNA. In the hippocampus of most healthy brains, hnRNP A3 was found to be present at moderate-to-high levels in the nuclei of both neurons and some glial cells, and at lower levels in the cytoplasm [49,70]. However, in FTLD cases with GGGGCC repeat expansions, the nuclear levels of hippocampal hnRNP A3 showed a significant reduction [49,50]. In the hippocampus of patients with *C9orf72* mutations, the decrease of nuclear hnRNP A3 was found to be associated with increased levels and aggregates of the dipeptide repeat protein poly-GA. Similar results have been reproduced in cellular models, such as the knockdown of nuclear hnRNP A3, leading to increases in repeat RNA, RNA foci, and DPRs (poly-GA, poly-GP, and poly-GR) in Hela cells and primary rat hippocampal neurons, and an increase in nuclear RNA foci in fibroblasts derived from *C9orf72* mutant patients [50]. In addition, in the granular layer of the dentate gyrus (DG), hnRNP A3 exhibited punctate neuronal cytoplasmic inclusions (NCIs), which showed co-positivity with p62 and accounted for 16.5–27.6% of p62-positive inclusions [49]. In the hippocampus of *C9orf72* mutation cases, p62-positive/TDP-43-negative punctate NCIs, star-like NCIs, and punctate neuronal intranuclear inclusions (NIIs) are some important pathological features [71]. Some subsets of these star-like NCIs were also found to show hnRNP A3 positivity [49]. In addition, hnRNP A3 punctate neuronal inclusions have also been found in the granule layer of the cerebellum in some cases with a *C9orf72* mutation [49,70]. Specifically, none of the above hnRNP A3 structures were found in non-*C9orf72*-mutant cases of FTLD or other related neurodegenerative diseases (e.g., ALS, AD, and Huntington’s disease). Interestingly, hnRNP A3 was also found in dystrophic neurites (DNs) both in *C9orf72*-mutant and a few non-mutant FTLD-TDP cases [49], which implies that there may be undiscovered non-*C9orf72*-related pathological mechanisms of hnRNP A3.

### 2.3. HnRNP C and D

Structurally, hnRNP C contains only one RRM, so its monomers must oligomerise into tetramers to form a complete functional unit for RNA interaction [72]. It was one of the first RNA-binding proteins to be identified as having a repressive effect on cryptic exon inclusion during splicing [31,73]. As a competitive regulator of the splicing factor U2AF65, the binding of hnRNP C to RNA can repress cryptic exon inclusion, while the depletion of hnRNP C allows U2AF65 to bind to RNA, resulting in an alternative splicing event in which the cryptic exon is included [31]. This repression of cryptic exons by hnRNP C may be even stronger than that of TDP-43 and FUS, as its depletion can result in greater cryptic exon alternative splicing events than the depletion of the other two proteins [31,74].

Immunoreactivity for hnRNP C was found in pathological inclusions in the frontal cortex and entorhinal cortex of FTLD-FUS cases [46], suggesting the possibility that hnRNP C has a synergistic effect with FUS in FTLD pathology. However, this aggregation in the inclusions was not accompanied by changes in total hnRNP C expression levels [46].

The function of hnRNP D is mainly related to RNA degradation. It can interact with AU-rich elements (AREs) that are involved in the regulation of gene expression and mRNA stability, thereby regulating ARE-directed mRNA degradation [75,76]. HnRNP D is one of the hnRNPs of which the expression can be autoregulated by the non-sense-mediated decay (NMD)-sensitive isoform pathway via negative feedback [77].

Similar to hnRNP A1 and C, hnRNP D deposits were also found in the inclusions of FTLD-FUS cases. In particular, in the frontal cortex and entorhinal cortex, hnRNP D inclusions were found mainly in the neuronal cytoplasm and in dystrophic neurites where no FUS pathology was identified. In contrast to hnRNP C, the formation of hnRNP D aggregates was accompanied by a significant increase in its expression [46].

### 2.4. HnRNP E2

The hnRNP E1, E2 and K are the only three hnRNPs that contain K homology (KH) domains, and they are therefore sometimes combined and referred to as the hnRNP K family [45,52]. It is because these three KH structural domains can interact independently with the target RNA sequence that makes it possible for this family of hnRNPs to form RNA interactions with a high degree of complexity and specificity [45]. In recent years, a few studies have found some association between hnRNP E2 and TDP-43 in FTLD cases. In frontal and temporal cortices and the hippocampal dentate gyrus (DG), hnRNP E2 was found to be present in the nucleus and cytoplasm of most FTLD cases, although nearly 25% of cases also showed no hnRNP E2 staining in any brain regions. While their levels varied individually, there were no significant differences among the different FTLD subtypes or pathogenetic cohorts [51]. However, in some SD patients with FTLD-TDP C pathology, most of TDP-43 positive DNs in the frontal and temporal cortex, as well as NCIs in DG granule cells of the hippocampus also showed hnRNP E2 positivity [51]. However, unlike TDP-43, although in these cases hnRNP E2 formed cytoplasmic inclusions, its levels in the nuclei of the corresponding neurons or cells did not appear to be reduced [51]. Since both TDP-43 and hnRNP E2 can be incorporated into stress granules under stress by interacting with T-cell intracellular antigen-1 (TIA-1), their co-localisation is hypothesised to arise through this assembling process [51]. However, it remains unclear why such TDP-43 and hnRNP E2 co-localised inclusions do not appear in other FTLD-TDP subtypes. Furthermore, although hnRNP E2 inclusions were found in approximately 2/3 of FTLD-TDP C cases, the clinical and neuropathological presentation of these patients did not appear to be significantly different from the remaining 1/3 of the patients without hnRNP E2 inclusions [51]. These unexplored issues may be due to sample size limitations. A following study by another group using a larger sample size repeatedly identified the presence of TDP-43-positive hnRNP E2 inclusions in FTLD-TDP C, and extended the finding that such hnRNP E2 inclusions were also present in some subtype A cases [52]. These authors found that in the frontal cortex, these hnRNP E2 inclusions were present mainly as long DN inclusions in layer II of the neocortex. Meanwhile, some scattered inclusions were also found in the anterior horn of the thoracic spinal cord in two cases [52]. However, this study did not report the clinical subtype of these FTLD-TDP cases in which hnRNP E2 inclusions were found, therefore it is not clear if all these cases were also from SD subtypes. Therefore, current studies on hnRNP E2 in FTLD are largely restricted to histological discoveries and validation, and pathogenic mechanisms remain to be further explored.

### 2.5. HnRNP F and H1

HnRNP H1-3 and F are structurally similar and belong to the same hnRNP H/F subfamily [78]. Functionally, they are mainly associated with the regulation of alternative splicing and polyadenylation, thus playing an important regulatory role in protein diversity [79,80]. Their association in FTLD pathology has been identified mainly from cases related to *C9orf72* HREs. In the cerebellum of *C9orf72*-associated patients, a large number of C9 RNA foci (up to 70%) were found to exhibit co-localisation with hnRNP H and F by RNA fluorescence in situ hybridisation, suggesting that hnRNP H and F may have a synergistic effect in the formation of RNA foci or may increase their toxicity and susceptibility to aggregation [53]. In addition, hnRNP H1 and F were also found to interact with a *C9orf72* HRE-associated DPR (poly-PR) [81]. Moreover, one study showed more directly relevant evidence for the association between hnRNP H and FTLD/ALS, in which approximately 50% of the investigated sporadic FTLD and ALS cases presented elevated levels of insoluble hnRNP H in the brain, as well as splicing abnormalities in a spectrum of hnRNP H downstream regulatory targets [54].

### 2.6. HnRNP G

HnRNP G is a hnRNP encoded by the RBMX gene on the X chromosome, so it is also known as the RNA-binding motif protein X (RBMX) protein. Functionally, it is a core alternative splicing regulator of many genes, including the FTLD-tau-associated *MAPT* (microtubule-associated protein tau) gene. It was found to promote exon 10 skipping of *MAPT* by interacting with serine/arginine-rich splicing factor 4 (SRSF4), suggesting that its abnormalities may have an important role in FTLD-tau pathology [82].

In addition, pathological deposits of hnRNP G were also found in FTLD-FUS cases, although unlike hnRNP A1, these deposits were not co-localised with FUS inclusions. HnRNP G inclusions were not only found in dystrophic neurites, but were also the only hnRNP found in the granular cells of the dentate fascia in FTLD-FUS cases. Interestingly, the expression of hnRNP G was found to be significantly increased in FTLD-FUS cases [46]. Moreover, hnRNP G has also been found to have an important role in DNA damage response and homologous recombination, and its aberration may therefore lead to compromised DNA damage repair in neurons [83,84,85].

### 2.7. HnRNP I and L

A central function of hnRNP I is to bind CU tracts and loop out alternative exons, thereby conducting splicing regulation [86,87]. Since CU tracts are located in the polypyrimidine-rich region of RNA, hnRNP I is also widely known as polypyrimidine tract-binding protein 1 (PRBP1) [88]. The splicing regulatory role of hnRNP I may be closely linked to FTLD pathology. Indeed, the FTLD frontal cortex was found to be rich in cryptic events in downstream RNA targets of hnRNP I [55]. 

HnRNP L can interact with hnRNP I and potentially act synergistically in pre-RNA splicing [89]. Although the main role of hnRNP L is also to repress cryptic exons, unlike hnRNP I, hnRNP L binds primarily to the CA-rich region of the intron upstream of the exon to be repressed [90]. However, abnormalities in the splicing regulation of hnRNP L in FTLD cases have not yet been identified.

As with hnRNP C, D, and G, deposits of hnRNP I and L were also found in the inclusions of FTLD-FUS cases. Among them, the distribution of hnRNP I inclusions was very similar to that of hnRNP D, mainly in neuronal cytoplasm and dystrophic neurites. However, unlike hnRNP D, the expression of hnRNP L was not significantly increased in the frontal cortex of FTLD-FUS cases [46].

### 2.8. HnRNP K

HnRNP K is one of the most widely expressed hnRNPs and accordingly its expression has been found in a wide range of brain regions, including frontal and temporal cortices and the hippocampus [91,92]. 

HnRNP K has six functional domains, including three KH domains (KH1, 2, 3) responsible for RNA binding, two domains related to nuclear localisation and shuttling (NLS and KNS), and a unique K interactive (KI) domain. As another hnRNP that contains KH domains, it is distinguished from hnRNP E1 and E2 by this KI domain, which facilitates its interaction with numerous protein targets in a large network of interactomes [39,93]. Although currently the pathological effects of aberrant hnRNP K are mainly studied in relation to cancer [94,95,96,97], it is also known to play multiple roles in the brain. It is crucial in the post-transcriptional regulation of many important genes related to axogenesis [98], myelination [99], synaptic plasticity (in hippocampal neurons) [100], and the maintenance of ATP levels (under cellular stress) [101].

In the healthy brain, hnRNP K predominantly localises to the nuclei of neurons. However, recent studies in our laboratory found that in many FTLD cases, pyramidal neurons from layers III and IV of the frontal cortices exhibited an increased mislocalisation of hnRNP K from the nucleus to the cytoplasm [32]. This mislocalisation resulted in the depletion of hnRNP K in the nucleus and punctate accumulation in the cytoplasm (even in neurites). Moreover, such hnRNP K mislocalisation was widespread in different FTLD subtypes, including FTLD-TDP (A, B, and C), FTLD-Tau, and FTLD-ni (no inclusions, a rare subtype), as well as in ALS cases. Although such mislocalisation has also been found in aged healthy brains, cases diagnosed with FTLD-TDP A and FTLD-tau exhibit significantly more mislocalisation than age-matched controls [32]. Interestingly, although this mislocalisation was not found in other cortical layers, such as layers I and II of the frontal cortices (superficial layers) [32], significantly elevated levels of this hnRNP K mislocalisation were found in the neurons of the cerebellar dentate nucleus in FTLD-TDP A and AD patients [56]. In particular, by comparing data from the same FTLD-TDP A cases in both studies, a significant correlation was found between the frequency of hnRNP K mislocalisation in the frontal cortex and cerebellum, suggesting that this hnRNP K pathology is homogeneous in the same sample. More importantly, the punctate inclusions formed by the mislocalised hnRNP K did not co-localise with TDP-43, tau, or p62-positive inclusions [32], suggesting that such hnRNP K mislocalisation may be a unique pathology-related structure rather than a component of these classical inclusions. Furthermore, these hnRNP K inclusions did not co-localise with markers of mitochondria, autophagosomes or stress granules [32], suggesting that mislocalised hnRNP K in the cytoplasm does not accumulate on these cellular structures. Notably, the neurons containing hnRNP K mislocalisation in both studies were the largest neuron types in the brain region where they are located, but it is unclear whether neuronal size is related to the susceptibility to hnRNP K mislocalisation. For example, it is not clear whether neurons with higher metabolic or energy demands are more susceptible.

In contrast to hnRNP E2 [51], the total amount of hnRNP K did not appear to be altered in the cortex where hnRNP K mislocalisation occurs, but rather there was a decrease in the nuclear level and an increase in the cytoplasmic level. When this nuclear depletion was simulated in a cellular model by hnRNP K knockdown (siRNA), a series of transcriptional and splicing events were altered. In particular, altered splicing resulted in the incorporation of many cryptic exons into the target transcripts (including 49 novel ones). Moreover, one of the cryptic exons, which was increased substantially in the cellular model of hnRNP K knockdown, was also identified in several FTLD cases containing hnRNP K mislocalisation [32]. Similarly, another group also identified this mislocalisation in the pyramidal neurons of the motor cortex of several *C9orf72*-associated ALS/FTD patients [57]. In addition, the nucleus/cytoplasm ratio of hnRNP K was also found to have a >40% decrease in fibroblasts derived from *C9orf72*-associated ALS patients and a 30% decrease in spinal motor neurons differentiated from patient-derived iPSCs [57]. 

In a *C9orf72* zebrafish embryonic model exhibiting *C9orf72* RNA toxicity, the motor neurons showed abnormalities of reduced axon length and excessive branching [57]. The overexpression of hnRNP K was found to attenuate this *C9orf72* toxicity and reduce axonal abnormalities in motor neurons, while the knockdown of hnRNP K in wild-type embryos resulted in axonal abnormalities similar to those of the *C9orf72* toxicity model [57]. In a HEK cell model, it was found that the deletion of the NLS/KNS domain in hnRNP K can lead to its localisation, predominantly in the cytoplasm and nuclear depletion. In contrast, the deletion of all three KH domains results in an increase in their cytoplasmic levels, but is not accompanied by nuclear depletion. Interestingly, hnRNP K models (lacking NLS/KNS or all KH domains) were found to lose their potential to alleviate *C9orf72* RNA toxicity [57], which may suggest that the nuclear localisation/shuttling and RNA binding capabilities of hnRNP K are important for regulation in *C9orf72*-related neurodegeneration. Through the zebrafish embryo model and *C9orf72* ALS patient brains, ribonucleotide reductase regulatory subunit M2 (RRM2) was suggested to play an important role in linking hnRNP K mislocalisation and *C9orf72* toxicity [57]. RRM2 was found to be a downstream protein regulated by hnRNP K, which is involved in the catalysis of deoxyribonucleotide formation in DNA synthesis [57]. In *C9orf72*-associated ALS cases, the nuclear depletion of hnRNP K cells in pyramidal neurons in layer IV of the motor cortex (as well as in spinal cord motor neurons) was accompanied by reduced RRM2 levels, but enhanced nuclear translocation, which suggests that the dysregulation of RRM2 may be associated with the mislocalisation of hnRNP K. In Hela cells, the knockdown of hnRNP K can lead to a reduction in RRM2 levels, while induced DNA damage can lead to the nuclear translocation of RRM2. Interestingly, the knockdown of hnRNP K also led to the nuclear translocation of RRM2, as well as DNA damage. As RRM2 is involved in the regulation of DNA replication and damage repair [102], these results imply that hnRNP K nuclear depletion/cytoplasmic mislocalisation may lead to abnormal DNA damage repair function by dysregulating downstream RRM2 expression. This abnormal DNA damage repair, in turn, could potentially be the cause of neurodegeneration. As evidence from the other direction, an injection of additional hnRNP K or RRM2 mRNA was able to reduce DNA damage in the C9 RNA toxicity zebrafish embryo model [57]. This study provides some encouraging insights into the pathological association of hnRNP K mislocalisation and neurodegenerative diseases. However, these results still cannot provide a complete chain of direct evidence completely linking *C9orf72* RNA toxicity, hnRNP K nuclear depletion/mislocalisation, reduced RRM2 expression, disturbed DNA damage repair, and neuronal degeneration, especially in the context of FTLD pathology. It is hoped that more mechanistic studies on hnRNP K in the future will connect these fragmented pieces of evidence, and reveal the complete pathway underlying hnRNP K mislocalisation and neurodegeneration.

### 2.9. HnRNP Q and R

HnRNP Q and R are two structurally and functionally similar hnRNPs [103] that are primarily involved in pre-mRNA splicing [104] and may have a regulatory role in TDP-43-controlled mRNA splicing [59,66]. The expression of hnRNP R was found to be higher in the frontal and temporal cortices of FTLD-TDP A and C, as well as FTLD-FUS cases than in healthy age-matched controls [58]. More importantly, hnRNP R and Q were found to form inclusions in the frontal cortex and the granule cell layer of the hippocampal DG region in all investigated FTLD-FUS cases. These inclusions were dominated by NCIs and were also occasionally accompanied by NIIs. Among the FTLD-FUS subtypes, the Neuronal intermediate filament inclusion disease (NIFID) samples exhibited more inclusions of hnRNP R and Q than in the atypical FTLD-U (aFTLD-U) sample controls [58]. In the frontal cortex, these inclusions mainly exhibited crescent-shaped, bean-shaped, or Pick-like structures, and were found adjacent to the nucleus. NIIs within the nucleus, on the other hand, although not rare, were less commonly found than NCIs, which usually exhibited a rod-like morphology. In addition, some aggregations of hnRNP R and Q also formed DNs in the frontal cortex controls [58]. Similarly, in the hippocampal DG region, the morphology of hnRNP R and Q NCIs were mainly bean-like and Pick-like, and their location was also mainly adjacent to the nucleus. Occasionally, some NCIs may also present as crescent-shaped or vermiform and some rod-like NIIs can be found within the nuclei of the granule cells. Notably, inclusions of hnRNP R exhibited co-localisation with FUS in both the frontal cortex and the granular layer of the hippocampal DG controls [58].

### 2.10. HnRNP U

HnRNP U is a structurally unique hnRNP. It is the largest of all hnRNPs [105], but it has neither an RRM nor a KH, the classical RNA recognition and binding domains [106,107]. It relies mainly on an RGG box near its C-terminus to bind RNA, and since this RRG box is glycine-rich, hnRNP U mainly binds G/U-rich RNA and regulates its splicing. In addition, hnRNP U has some DNA and chromosome binding ability, as it has an SAP domain near its N-terminus. Due to this ability, it may also participate in a range of DNA regulatory pathways, such as regulating telomere length and DNA damage response [106,107].

In cellular models, hnRNP U was found to bind synergistically to wild-type or ALS-associated mutant TDP-43, FUS, and Ataxin 2 [59,60,61]. Moreover, in yeast models, hnRNP U was found to bind synergistically with ubiquilin-2 as a binding partner, similar to hnRNPs A1 and A3 [62]. Possibly due to being a binding partner of TDP-43, hnRNP U was discovered to mediate the neurotoxicity related to TDP-43 overexpression in the NSC-34 cell model [63]. Although this was the only evidence from cellular models, it suggests the existence of synergistic regulation between different hnRNPs. This co-regulation may synergistically amplify the toxicity and pathological effects of aberrant hnRNPs. In addition, hnRNP U expression was found to be elevated in the frontal cortex of FTLD-FUS cases, along with hnRNPs A2/B1 and D [46].

## 3. Conclusions and Discussion

In recent years, a growing number of studies have identified an association between hnRNPs and FTLD, as well as other neurodegenerative diseases. Although distinct hnRNPs may appear in diverse types of neurons or in cells of different brain regions, overall, the majority of hnRNPs associated with FTLD show nuclear to cytoplasmic translocation, at times accompanied by the formation of cytoplasmic inclusions. Most current research related to these lesser studied hnRNPs (apart from TDP-43 and FUS) is restricted to the identification of abnormal cellular localisation in FTLD cases. These studies have provided some valuable evidence for the pathological analysis of hnRNPs and laid the foundation for further mechanistic studies. A subset of studies, such as those on hnRNPs A1 and K, have begun to explore the mechanisms behind the association of hnRNPs with FTLD using both cellular and animal models. However, due to the diversity of hnRNPs and the complexity of FTLD-related neurodegeneration, the pathology of FTLD may be mediated by a series of inter-related abnormalities in RNA metabolism and regulation, which cannot be attributed to the functional abnormalities of a single protein. Taken together, the hypothesis of one pathway linking hnRNP pathologies and FTLD may be proposed (Figure 1). Cellular stress induces the formation of stress granules which contain hnRNPs. The homologous or sometimes even heterologous hnRNPs are able to bind via their own low-complexity domains (prion-like sequences), which leads to their LLPS and further promotes granule formation. As these granules mature, they gradually fibrilise and form amyloid-like aggregates, which in turn leads to the formation of inclusions. This process may manifest itself as a mislocalisation of hnRNPs in the cytoplasm and a decrease in the ratio of their nuclear/cytoplasmic fraction. This deficiency of hnRNPs in the nucleus can lead to abnormalities or a loss of function in the regulation of downstream RNA targets. Such regulatory abnormalities may lead to aberrant splicing, resulting in the NMD degradation of target mRNA (NMD-sensitive transcripts), premature termination (NMD-insensitive transcripts), or the incorrect translation of critical functional proteins (transcripts not containing PTC). Reduced levels of functional proteins lead to abnormalities in neuronal function, such as the reduced axonal growth or the inability to maintain normal DNA damage repair, which allows cellular stress to accumulate and could lead to neurodegeneration. During these processes, the autoregulation of hnRNPs via NMD and/or other pathways may further exacerbate this vicious cycle (Figure 1).

However, there is currently insufficient evidence to systematically support and validate all the steps in this hypothesis. More importantly, the current evidence is mostly based upon histological studies looking at associations between hnRNPs and FTLD pathology, but there is no direct evidence to determine the cause and effect. Furthermore, the current pathology of hnRNPs is mostly considered to be related to the nuclear depletion and loss of function, yet it is unknown whether the formation of aggregates and inclusions can also result in an abnormal gain of functions. In addition, there is a compensatory effect between many functionally similar proteins [108] and this phenomenon may also exist between hnRNPs [79]. This compensatory effect may mask the depleting effects of some hnRNPs and increase the complexity of mechanistic studies. Therefore, more studies for the underlying mechanisms between hnRNP pathologies and FTLD are needed in the future.

In addition, some studies have also implied that there may be pathological mutations in hnRNPs which have not yet been explored. It might be feasible for these mutations to cause abnormalities in the nuclear localisation of hnRNPs and enhance their translocation to the cytoplasm. Perhaps GWAS analysis on multiple central and large-scale data can help identify some potential risk variants of hnRNPs. On the other hand, it might be possible to investigate which key downstream RNAs are expressed abnormally due to nuclear depletion/cytoplasmic mislocalisation of hnRNPs via high-throughput RNA-seq or in situ hybridisation. If the upstream and downstream mechanisms associated with the pathology of hnRNPs can be identified, it might be possible to discover potential targets for intervention, thereby offering new possibilities for the prevention and treatment of FTLD.

Interestingly, the occurrence of hnRNP inclusions appears to be neuron-subtype-specific, which suggests that hnRNP pathologies possess selective neuronal vulnerability. For example, hnRNP K inclusions are present in pyramidal neurons [32], whilst TDP-43 inclusions tend to occur in distinct cortical layers in the different FTLD-TDP pathological subtypes [11,109]. Selective neuronal vulnerability is a pathological feature of many neurodegenerative diseases. For example, in AD, pyramidal neurons in the entorhinal cortex and hippocampus are more vulnerable to tau pathology, while in Parkinson’s disease, dopaminergic neurons in the substantia nigra are more vulnerable to alpha-synuclein pathology [110]. Several studies have now found that the most vulnerable neurons in FTLD cases are the von Economo neurons (VENs), fork cells, and Betz cells in the frontoinsular, anterior cingulate cortex and midcingulate cortex [110,111,112,113]. These findings were mainly identified from cases associated with TDP-43 pathology. Although TDP-43 is a member of the hnRNP family and many other hnRNPs have synergistic roles in the pathology of TDP-43, it is unclear whether the pathologies of other hnRNPs exhibit selective vulnerability in different types of neurons. Moreover, the mechanisms behind these selective vulnerabilities also remain unknown, although it has been suggested that they may be related to a number of factors such as genetics, protein homeostasis, mitochondrial energy supply, and calcium homeostasis [110]. Further identification of susceptible neuron types corresponding to different hnRNPs and joint analysis with the genetic, expression, distribution, structural and functional characteristics of different hnRNPs may provide insights into the mechanisms of selective vulnerability.

Current research on the relationship between hnRNP pathologies and FTLD still faces many limitations. Most studies are currently limited to protein analysis (such as immunohistochemistry and Western blotting) in post-mortem tissues, which may only highlight the end stage of the disease. In post-mortem studies, the presence of proteinaceous inclusions or aberrant cellular localisation of a protein is indicative that hnRNPs may play a role in disease, as is clear from TDP-43 and FUS studies over the years. It is worth noting that other hnRNPs might not be forming cytoplasmic aggregates or inclusions, but their function and regulatory roles might still be compromised in the context of disease. Additionally, post-mortem studies make it difficult to study the progression of the pathologies or facilitate the identification of biomarkers at early stages. Techniques such as single-cell RNA-seq may shed some further insight into how the hnRNP network functions together in the context of an FTLD brain, and may further shed light into specific neuronal vulnerability. Additionally, further hnRNP pathology studies may require the generation of suitable models for mechanistic insights.

Recently, downstream splicing targets of TDP-43 have been investigated for potential therapeutic avenues. In particular, therapeutic attempts to target and rescue full-length *STMN2* expression have been successful both in vitro and in vivo. Abnormal axonal regeneration and cell death due to TDP-43 depletion were rescued by increasing *STMN2* expression in a human motor neuron model derived from induced pluripotent stem cells [24,25], and the virus-mediated reintroduction of human *STMN2* expression in a *Stmn2*-knockout mouse model was able to rescue motor deficits [114]. Less than 5 years after the publication of these findings, a Phase 1 clinical trial has been initiated targeting the upregulation of *STMN2* expression in ALS patients (ClinicalTrials.gov Identifier: NCT05633459), which is a breakthrough in therapeutic attempts to target genes downstream of hnRNP (TDP-43). Similarly, the loss of nuclear TDP-43 leads to cryptic exon inclusion in downstream UNC13a, resulting in the reduced expression of the synaptic protein. The presence of UNC13a SNPs further exacerbate cryptic exon inclusion upon TDP-43 dysfunction. UNC13a SNPs rs12608932 and rs12973192 are now recognised as potential risk variants for ALS/FTD, and these findings have therefore introduced additional potential biomarkers and therapeutic targets. Currently, lithium carbonate is being trialled for the treatment of ALS symptoms in a subgroup of patients homozygous for the C-allele at SNP rs12608932 in UNC13A [115].

Moreover, downstream splicing events caused by aberrant hnRNPs other than TDP-43 have also been found to have potential value as biomarkers and therapeutic targets. For example, a recent study has shown that the splicing events leading to the inclusion of the cryptic exon FAM160B2 are significantly upregulated in the brains of patients with high levels of hnRNP K mislocalisation [32], which could potentially be a marker for hnRNP K abnormalities, though functional consequences of this mislocalisation are still currently unknown. In addition, another recent study showed that in a *C9orf72* zebrafish model, RNA toxicity of C9 could be rescued by replenishing the expression of hnRNP K or its downstream target *RRM2* [57]. Therefore, hnRNPs have great research value in both novel therapeutic approaches and diagnostic tools. Although research on hnRNPs is still in its early stages, breakthroughs might revolutionise the research and treatment of FTD and related neurodegenerative diseases.

In conclusion, an increasing number of studies in recent years have found hnRNPs to be associated with FTLD. This has contributed to an elevated focus on hnRNPs in the field of FTD and FTLD research. Several studies have already begun to explore whether hnRNPs are causally related to FTLD, and have provided staged evidence for the mechanisms behind this relationship. A few studies have even attempted to alleviate neurotoxicity and neurodegeneration by intervening in abnormal splicing due to mislocalised hnRNPs or their downstream targets in cellular and animal models, as well as clinical trials. These findings in recent years have shed light on the future studies for the pathology of hnRNPs in FTLD and the related neurodegenerative diseases.

Under cellular stress, hnRNPs are able to form into stress granules by liquid–liquid phase separation to regulate protein translation towards promoting survival. Mutant hnRNPs exacerbate the recruitment of hnRNPs to stress granules, causing them to lose their reversibility and mature. Mature stress granules, accumulated with large amounts of hnRNPs and RNA, form insoluble amyloid-like fibrils and aggregates. The accumulation of hnRNPs in the inclusions causes their nuclear depletion, which leads to a series of splicing abnormalities, including cryptic exon inclusion and exon skipping. Most of the aberrant NMD-sensitive transcripts are degraded via the NMD pathway, while the remaining aberrant mRNAs are translated into abnormal proteins or have reduced protein levels due to the premature termination of translation. Some of these abnormal or deficient proteins lead to compromised DNA damage repair and others lead to abnormal neuronal function such as abnormal axonal growth. This would lead to further accumulation of cellular stress, promoting the maturation of stress granules, while the accumulation of hnRNPs in the nuclear and/or cytoplasmic inclusions could downregulate their expression through negative feedback autoregulation. However, this autoregulatory effect through the upregulation of NMD-sensitive isoforms would simultaneously decrease the expression of other proteins. Together, these processes lead to a vicious cycle and eventually result in neurodegeneration. Created with BioRender.com.

## Figures and Tables

**Figure 1 cells-12-01633-f001:**
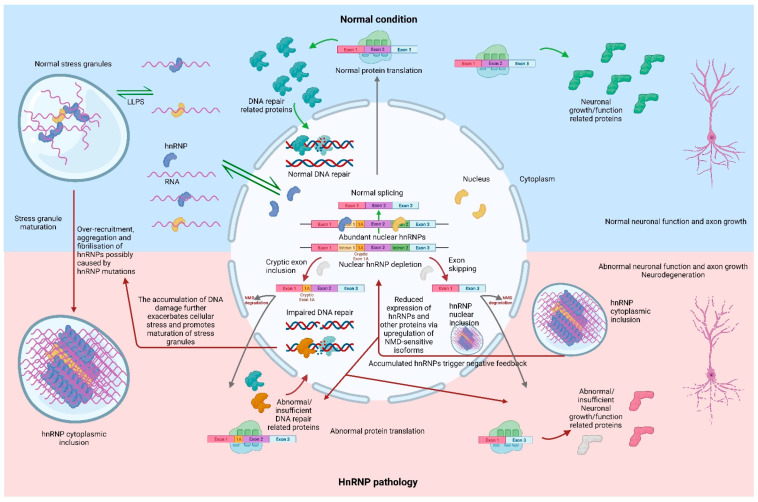
The schematic representation of a hypothetical pathway in which hnRNP pathologies are associated with FTLD. Created with BioRender.com.

**Table 1 cells-12-01633-t001:** Structural information and current pathological findings on hnRNPs (other than TDP-43 and FUS) associated with FTLD.

Name	Other Names	Amino Acid Length	RNA Recognition Domains	DNA-Binding Domains	Nuclear Localisation/ Transport Domains	Current Representative Pathological Findings Related to FTLD
hnRNP A1		372	2 RRMs, 1 RGG box		1 M9	Nuclear depletion and cytoplasmic aggregation (FTLD-FUS cases [46]) Bind to TDP-43 and increase more aggregation-prone hnRNP A1B (cell models [47]) PARylation of hnRNP A1 may be necessary for its translocation and toxicity, while the binding of PAR or PARylated proteins is important for its recruitment to stress granules (cell models [48])
hnRNP A2B1		353	2 RRMs		1 M9, 1 NLS	Increased expression in FTLD-FUS cases [46]
hnRNP A3		378	2 RRMs			Punctate neuronal cytoplasmic inclusions co-positivity with p62 in multiple brain regions (C9 FTLD cases [49]) Hippocampal hnRNP A3 reduction in C9 FTLD cases [49,50] Decrease in nuclear hnRNP A3 associated with increased levels and aggregates of poly-GA (cell models [50]) Knockdown of nuclear hnRNP A3 leading to increases in repeat RNA, RNA foci, and DPRs (cell models [50])
hnRNP C		306	1 RRM		1 NLS	Positivity for hnRNP C in pathological inclusions in the frontal cortex and entorhinal cortex, but no hnRNP C expression changes (FTLD-FUS case [46])
hnRNP D	AUF1, LaAUF1, JKT41-binding protein	355	2 RRMs			Positivity for hnRNP D in pathological inclusions in the frontal cortex and entorhinal cortex with increased hnRNP D expression (FTLD-FUS case [46])
hnRNP E2	PCBP2, Alpha-CP2	365	3 KHs			HnRNP E2 cytoplasmic inclusions (co-localising with TDP-43) in multiple brain regions of SD patients with FTLD-TDP C and some FTLD-TDP A cases [51,52] Although cytoplasmic inclusions are formed, hnRNP E2 levels in the nuclei of the corresponding neurons or cells did not appear to be reduced [51]
HnRNP F	nucleolin-like protein mcs94-1	415	3 RRMs			In the cerebellum of patients with *C9orf72* mutations, many C9 RNA foci (up to 70%) exhibited co-localisation with hnRNP H [53]
hnRNP G	RNA-binding motif protein, X chromosome (RBMX), Glycoprotein p43	391	1 RRM, 1 unspecified domain		1 NLS	Deposits of hnRNP G were found in the inclusions of FTLD-FUS cases, not only in dystrophic neurites, but also in the granular cells of the dentate fascia (FTLD-FUS cases [46]) Increased expression of hnRNP G was found in these brain regions (FTLD-FUS cases [46])
HnRNP H1		449	3 RRMs			Elevated levels of insoluble hnRNP H in the brain, as well as splicing abnormalities in a spectrum of downstream hnRNP H regulatory targets (sporadic FTLD and ALS cases [54])
hnRNP I	PTB, PPTB-1	531	4 RRMs			Positivity for hnRNP I in pathological inclusions in the frontal cortex and entorhinal cortex with no changes of hnRNP L expression (FTLD-FUS case [46]) RNA from the frontal cortex of FTLD cases was found to contain an increased number of cryptic exons in the targets of hnRNP I [55]
hnRNP K	TUNP	463	3 KHs, 1 KI (containing 5 RRG boxes)			Pyramidal neurons from layers III and IV of the frontal cortices and neurons in the cerebellar dentate nucleus showed increased nucleus-to-cytoplasm mislocalisation of hnRNP K (no total hnRNP K level changes), resulting in the nuclear depletion of hnRNP K and punctate accumulation in the cytoplasm and even in neurites (FTLD cases [32,56]) Mislocalised hnRNP K did not co-localise with TDP-43, tau, or p62-positive inclusions. HnRNP K cytoplasmic puncta also do not co-localise with markers of mitochondria, autophagosomes or stress granules (FTLD cases [32]) HnRNP K-knockdown induced altered transcriptional and splicing events, particularly, cryptic exon inclusion events (cell model and FTLD cases [32]) Deletion of the NLS/KNS domain in hnRNP K can lead to its localisation predominantly in the cytoplasm and nuclear depletion (cell model [57]) Overexpression of hnRNP K-attenuated C9 toxicity and reduced axonal abnormalities in motor neurons; knockdown of hnRNP K resulted in axonal abnormalities similar to those caused by C9 toxicity (Zebrafish embryo model [57]) HnRNP K may regulate and maintain DNA damage repair via regulating RRM2 expression, while abnormalities in hnRNP K may lead to compromised DNA damage repair and the accumulation of DNA damage (zebrafish embryo and cell model [57])
hnRNP L	SRRF	589	4 RRMs			Positivity for hnRNP L in pathological inclusions in the frontal cortex and entorhinal cortex with no changes in hnRNP L expression (FTLD-FUS case [46])
hnRNP Q	SYNCRIP, CRY-RBP, NS1-associated protein	623	3 RRMs		1 NLS	HnRNP Q formed inclusions in the frontal cortex and the granule cell layer of the hippocampal DG region in FTLD-FUS cases [58]
hnRNP R		633	3 RRMs, 1 RRG box		1 NLS	Expression of hnRNP R was higher in the frontotemporal cortex of FTLD-TDP A and C, as well as FTLD-FUS cases [58] HnRNP R formed inclusions in the frontal cortex and the granule cell layer of the hippocampal DG region in FTLD-FUS cases [58] Inclusions of hnRNP R exhibited co-localisation with FUS in both the frontal cortex and the granular layer of hippocampal DG controls [58]
hnRNP U	GRIP120, SAF-A, nuclear p120 ribonucleoprotein	825	1 RRG box	1 SAP		HnRNP U expression was found to be elevated in the frontal cortex of FTLD-FUS cases, along with hnRNP A2/B1 and D [46] HnRNP U was found to bind synergistically to wild-type or ALS-associated mutant TDP-43, FUS, and Ataxin 2 as binding partners (cell model [59,60,61]) HnRNP U was found to bind synergistically with ubiquilin-2 as a binding partner, similar to hnRNP A1 and A3 (yeast model [62]) HnRNP U was discovered to mediate the neurotoxicity related to TDP-43 overexpression in the NSC34 cell model [63]

## Data Availability

Not applicable.

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
