# Peer review of "HnRNP Pathologies in Frontotemporal Lobar Degeneration"

_cells, 2023, doi:10.3390/cells12121633_

Round 1

Reviewer 1 Report

This is a very interesting manuscript regarding FTD. It explore more complex and not well known molecular mechanisms involved not only in the pathogenesis but also useful to explain the extensive phenotype variability.

 On the other hand, the confirmation of the hypothesis suggested by the authors, might contribute to explore new therapeutic pathways and actually to explain the recent and unsuccessful therapeutic approaches  in the neurodegenerative diseases

Author Response

Thank you for your comments, please see the responses in the document attached.

Reviewer 2 Report

The review summarize the role of the heterogeneous ribonucleic acid protein (hnRNP) family other than TDP-43 and FUS in frontotemporal lobar degeneration. The review is comprehensive and well written

Comments

-The conclusion should be improved to describe how research can be applied to this field. Should hnRNP be studied as diagnostic markers or target for therapy or both

The English language needs minor editing

Author Response

(The authors gave the same response as above.)

Reviewer 3 Report

I find this review manuscript unsuitable for publication as the same group has published very similar reviews in Acta Neuropathologica in 2020 and Molecular Neurobiology in 2021. The 2020 review in particular is set up almost exactly the same, by HnRNP protein, and may actually be more comprehensive.

Author Response

(The authors gave the same response as above.)

Round 2

Reviewer 3 Report

While I still believe this review is too similar to previous reviews, the authors point out there are some small differences with some updated data, so I will recommend for publication.